# Feasibility of Muscle Endurance Testing in Critically Ill Trauma Patients: A Pilot Study

**DOI:** 10.3390/healthcare11010053

**Published:** 2022-12-24

**Authors:** Sun Hyun Kim, Ho Jeong Shin, Myung-Jun Shin, Myung Hun Jang

**Affiliations:** 1Department of Trauma Surgery and Surgical Critical Care, Regional Trauma Center, Busan 49241, Republic of Korea; 2Department of Physical Therapy, Graduate School, Catholic University of Pusan, Busan 46252, Republic of Korea; 3Department of Rehabilitation Medicine, Pusan National University Hospital, Busan 49241, Republic of Korea; 4Biomedical Research Institute, Pusan National University Hospital, Busan 49241, Republic of Korea

**Keywords:** functional index, muscle strength, physical endurance, intensive care unit-acquired weakness, surface electromyography

## Abstract

Intensive care unit-acquired weakness (ICUAW) occurs secondary to patients treated for life-threatening conditions in the ICU being diagnosed based on the Medical Research Council sum score (MRC-SS). However, patients often complain of fatigability and poor endurance, which are not evaluated by muscle strength. In this study, we explored the feasibility of assessing muscle quality and endurance in trauma ICU patients. The modified Functional Index-2 (FI2) testing was applied to evaluate muscle endurance. The maximal voluntary contraction (MVC) was measured when evaluating the MRC-SS using surface electromyography (sEMG), and the fatigue index (FI) was also recorded at the time of endurance testing. The ultrasonic muscle echogenicity by gray-scale analysis of rectus femoris (RF) and tibialis anterior (TA) muscles was evaluated at the initial (<72 h) and end of ICU care. A total of 14 patients were enrolled in this study. Fatigue was induced in eight patients (fatigue group), and six (non-fatigue group) completed endurance testing. All patients except one had an MRC-SS exceeding 48 points. There was no difference in US echogenicity, MRC-SS, and FI between groups. In sEMG, the root mean square (RMS) values of MVC in RF and TA muscles showed a significant difference (*p* < 0.05). To evaluate and predict the functional activity of ICU patients, measuring muscle strength alone is insufficient, and it is necessary to evaluate muscle endurance. In this respect, the modified FI2 test and sEMG monitoring are considered to be promising procedures for evaluating the muscle condition of critically ill patients even in complex situations in the ICU.

## 1. Introduction

Intensive care unit-acquired weakness (ICUAW), which occurs secondary to patients treated for life-threatening conditions in the ICU, occurs in approximately 25% to 100% of ICU inpatients [1]. Moreover, it is associated with high morbidity and mortality of acute critical illnesses, as well as functional disability after ICU discharge [2,3]. A clinical diagnosis of ICUAW can be made through muscle strength assessment at the bedside, and for muscle strength assessment, the Medical Research Council sum score (MRC-SS), which evaluates the strength of muscle groups in the upper and lower limbs, can be used as a clinical diagnostic tool. ICUAW is diagnosed as the MRC-SS being <48 out of 60 [1]. Because the MRC-SS has a ceiling effect, muscle endurance and functional assessments are needed to know the patient’s actual functional level. In addition to the MRC-SS, physical functions such as rolling, supine to sit, sitting, and standing can be evaluated in ICU. Muscle strength did not reflect the patient’s fatigability, and few studies have evaluated endurance as an indicator of muscle quality in the ICU [4].

Prolonged immobilization and bed rest after trauma can cause muscle atrophy and weakness which can lead to a decrease in functional mobility [5]. However, functional assessment in trauma patients has many limitations due to specific conditions related to injuries and treatment. In particular, fractures of the spine, pelvis, and lower extremities are major factors limiting out-of-bed activities in the acute stage. For this reason, there are no validated functional evaluation items for trauma patients. Herein, we explored the feasibility of assessing muscle quality and endurance to evaluate and predict the functional activity of trauma ICU patients.

## 2. Materials and Methods

### 2.1. Study Design

We performed a prospective study from December 2019 to December 2020 of critically ill patients who were admitted for chest and abdominal injuries at the Regional Trauma Center in Pusan National University Hospital. Based on the ICU early mobilization protocol, patients who satisfied the safety criteria were included (Appendix A). Patients with an appropriate level of consciousness (−2 < Richmond Agitation–Sedation Scale < 2) and good cooperation (standardized five questions: 5) were identified for reliable protocol performance. The ultrasonic muscle echogenicity by gray-scale analysis of rectus femoris (RF) and tibialis anterior (TA) muscles was evaluated at the initial ICU admission (<72 h) and at the discharge from ICU. When ICU discharge was determined, the MRC-SS calculation and endurance testing were performed. In addition, the maximal voluntary contraction (MVC) and the fatigue index (FI) of each muscle were recorded using surface electromyography (sEMG). The hospital Institutional Review Board approved the study (IRB number: 1904-005-078).

### 2.2. Ultrasound

Ultrasound was performed by one of the authors (MHJ) with 5 years of ultrasound experience. B-mode images were acquired using a 10 MHz linear wireless portable probe (SONON 300L; Healcerion Co., Ltd., Seoul, Republic of Korea). Standardized depth and gain in the same image were preset for all patients. Muscle echogenicity of RF and TA muscles was analyzed using the histogram function of Image J software (https://imagej.nih.gov/ij/index.html NIH, Bethesda, MD, USA, accessed on 15 January 2021) [6].

### 2.3. Muscle Strength

For the patient’s muscle strength, the MRC-SS was used, which was the sum of the values of each 12-muscle group. The MRC-SS ranges from 0 to 60 points [1]. In the supine position, the maximal isometric contraction was measured with the head end of the bed raised by 10° [7].

### 2.4. Modified FI2 Endurance Testing

The Functional Index-2 (FI2) testing for patients with myositis was modified and performed as endurance testing for ICU patients (Figure 1). The modified FI2 test repeats two standardized movements for knee extension (40 BPM, 60 repetitions) and ankle dorsiflexion (80 BPM, 120 repetitions) at a constant pace using a metronome for 3 min and counts the number of repetitions. During trials, the task was stopped if the correct movement was not performed more than three repetitions, such as an inability to repeat at a constant pace to the metronome due to muscle fatigue, or active range of motion (ROM) does not reach the passive ROM [8]. At the end of each task, the patient’s muscular exertion was scored on the Borg Category Ratio-10 scale. The modified FI2 score for each exercise was calculated as the percentage of the maximum number of repetitions and divided by 10 (0–10) [4]. Patients who completed the 3 min task were assigned to the non-fatigue group, and patients who did not complete knee extension or ankle dorsiflexion were assigned to the fatigue group.

### 2.5. Surface EMG

EMG signals were acquired using a Bluetooth sEMG device (MOT10; PhysioLab Co., Ltd., Busan, Republic of Korea) on RF and TA muscles. The interelectrode distance, electrode placement procedure, and skin preparation followed the standard Surface Electromyography for the Non-Invasive Assessment of Muscles guidelines [9]. The MVC was measured before the modified FI2 test, and the FI of each muscle was simultaneously recorded during endurance testing. Further, the average value of %MVC for each muscle contraction was analyzed. The root mean square (RMS) value was used to analyze and process the recorded EMG signal (Figure 2).

### 2.6. Statistical Analysis

SPSS (ver. 18.0 for Windows, SPSS Inc., Chicago, IL, USA) was used for analysis. Continuous variables were compared between groups using the Mann–Whitney U test. The level of statistical significance was *p* < 0.05.

## 3. Results

Fourteen patients were included in the study. The fatigue group comprised eight people (mean age, 49.3 ± 21.6 years), and the non-fatigue group comprised six people (mean age, 53.2 ± 5.8 years). There were no statistically significant differences in injury severity scale (ISS), length of ICU stays, mechanical ventilator days, grip strength of the dominant hand, and MRC-SS between groups (Table 1). Only one patient’s MRC-SS was <48 in the fatigue group, which was the criterion for the ICUAW.

The modified FI2 score of the fatigue group was rated as 5.78 ± 3.3 in RF and 5.45 ± 9.8 in TA, showing a performance rate of less than 60%. The RMS values of MVC in RF (*p* = 0.048) and TA (*p* = 0.001) evaluated during strength evaluation showed statistically significant differences between the two groups. However, there was no difference between FI and the RMS values of %MVC recorded during the task. In addition, there was no significant difference in changes of muscle echogenicity during ICU admission (Table 2).

## 4. Discussion

Skeletal muscle wasting occurs from the early acute phase of ICU care; however, there is no standardized procedure for evaluating muscle quantity and quality status. Similar to chronic obstructive pulmonary disease, muscle fiber-type shifting from type 1 to type 2 in limb muscles also occurs in critically ill patients. In addition, myosin filaments loss and sarcomere structural damage also contribute to muscular weakness [10]. These changes lead to a decrease in muscle endurance, which is closely related to the impairment of functional activity [11].

Muscle strength can be measured using a dynamometer, tensiometer, or manual muscle testing, and to evaluate muscle mass, magnetic resonance, ultrasound, computed tomography, and dual-energy X-ray absorptiometry can be used [12]. However, there are limitations in predicting a patient’s prognosis or functional recovery with only one evaluation tool. Additionally, assessment of the functional level can predict the patient’s functional recovery and prognosis after discharge from the ICU [13]. Functional measurement tools in the ICU include the functional status score for the intensive care unit (FSS-ICU), activity measure for post-acute care (AM-PAC), Chelsea critical care physical assessment tool (CPAx), Johns Hopkins highest level of mobility scale (JH-HLM) and others. However, there are situations in which weight-bearing tasks such as standing and walking are impossible in some trauma patients, and there are no validated functional evaluation tools for ICU trauma patients.

This pilot study showed the applicability of endurance assessment in critically ill patients. Our data showed that modified FI2 could assess muscle function in patients with high muscle strength. In addition, measuring MVC with sEMG may help predict fatigability as an adjunct of the MRC-SS (Figure 3).

Herein, there was no difference in the MRC scale between the patient groups; however, the RMS values of MVC in RF and TA muscles showed significant differences. The fatigue task could not be completed in the low-RMS amplitude group. Watanabe et al. reported a correlation between muscle strength and RMS in the elderly [14]. Similarly to the elderly, critically ill patients develop muscle atrophy and weakness and show the characteristics of neuromuscular weakness, such as critical illness polyneuropathy (CIP) and critical illness myopathy (CIM) [15]. Loss of functional motor units with axonal injury in CIP appears as a reduced recruitment interference pattern in needle EMG. In CIM, the amplitude is reduced due to the reduced number of functional muscle fibers [16]. Changes in the pool of the motor unit during critical care may be associated with a decrease in EMG amplitude.

Amici et al. showed the possibility of FI2 being useful not only in myositis but also in other neuromuscular diseases. The modification was essential to apply the dynamic FI2 test to trauma patients in the ICU. The modified FI2 had an exercise intensity of about 40%MVC, and muscle fatigue was less than 10% in FI. However, endurance testing was not completed even in patients with high muscle strength (MRC-SS ≥ 48) in the fatigue group. Previous studies have also revealed an imbalance between endurance and maximal strength in neuromuscular diseases such as metabolic myopathies and myositis. Particularly in ICU patients with high MRC scores, it will be possible to evaluate the measurable functional status and detect subtle changes in recovery during early rehabilitation. This is because the modified FI2 is much more variable than the MRC-SS.

Recent quantitative ultrasound can show early pathologic changes in the muscle in critical illness, which may provide early diagnosis and management. In addition, ultrasound offers the advantage of being non-invasive and portable and without ionizing radiation [17]. Although some studies showed significant changes in gray-scale imaging or muscle thickness in the early course of ICUAW, this was not sufficient to ensure ultrasound reliability [17,18]. In our study, significant changes in muscle echogenicity during critical care were not identified in both groups. The main subjects of the previous ultrasound study were acute respiratory distress syndrome and severe sepsis, but the trauma patients in this study were not in systemic inflammation.

This study has some limitations. We modified and applied two subcomponents of the FI2 task; however, we could not secure validity and reliability. In the ICU environment, it was difficult to perform repetitive exercises to evaluate upper extremity muscle endurance due to central venous catheters, arterial catheters, intravenous lines, and various monitoring devices such as electrocardiograms that limit patient activity. Therefore, we devised a method for evaluating the lower extremities in the supine position. In addition, for objective evaluation, we evaluated only patients with a normal state of consciousness (−2 < Richmond Agitation–Sedation Scale < 2) and good cooperation (standardized five questions: 5). Our study enrolled a small number of participants, limiting the strength of our conclusion. The data collected in this study cannot allow us to conclude that the endurance test we designed is feasible in ICU. Despite the small sample, it is meaningful that the test can be performed in a limited ICU environment. Therefore, based on this result, more patients should be studied in further research. Further research is needed using objective diagnostic tools such as electrodiagnostic studies to determine the ICUAW of critically ill patients.

This pilot study suggested the possibility of a muscle endurance test to assess muscle quality in trauma ICU patients. Screening ICU patients with high muscle strength but reduced endurance may provide another criterion for the need for early mobilization.

## Figures and Tables

**Figure 1 healthcare-11-00053-f001:**
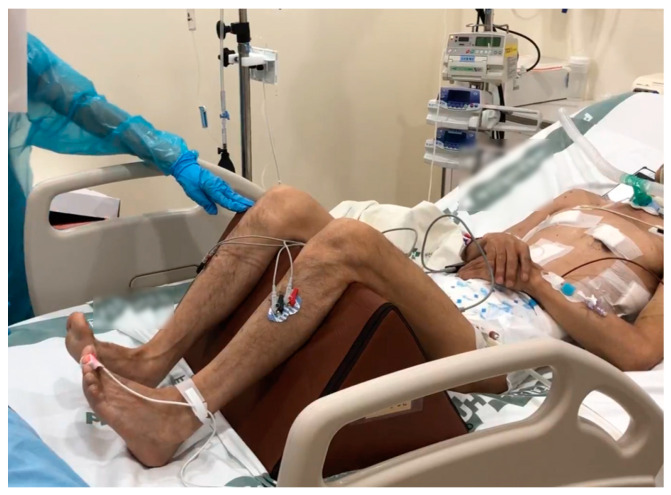
Modified FI2 endurance task using surface electromyography in the ICU patient.

**Figure 2 healthcare-11-00053-f002:**
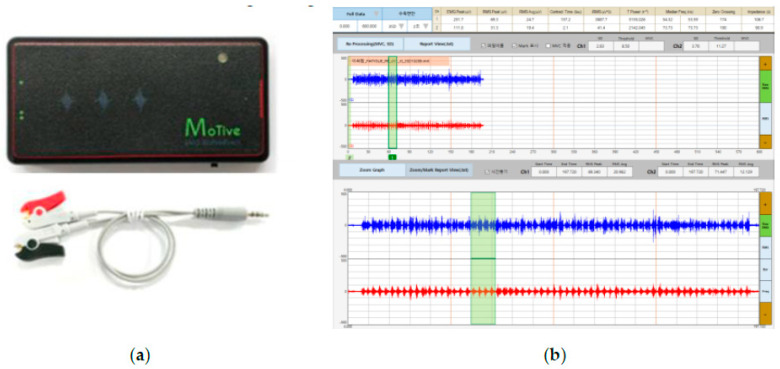
(**a**) Bluetooth surface electromyography device (MOT10; PhysioLab Co., Busan, Republic of Korea); (**b**) user interface of surface electromyography software (MoTive-Rs v1.0).

**Figure 3 healthcare-11-00053-f003:**
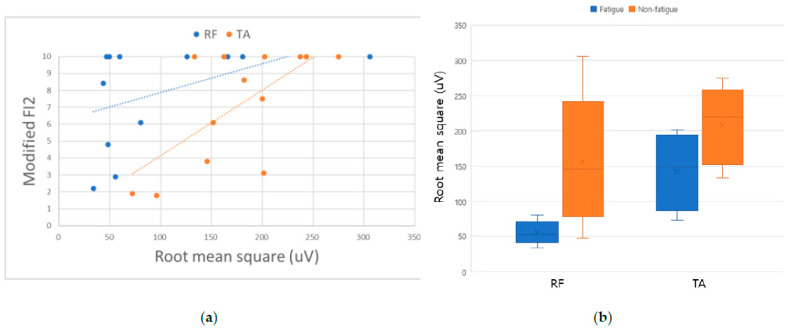
(**a**) The relationships with modified FI2 endurance testing and RMS of MVC in ICU patients. Positive linear correlations were observed between modified FI2 and RMS of MVC (Spearman’s rho: 0.618 in RF and 0.631 in TA); (**b**) comparison of RMS values obtained with MVC in RF and TA in the two groups.

**Table 1 healthcare-11-00053-t001:** Characteristics of patients.

	Fatigue Patients (n = 8)		Non-Fatigue Patients (n = 6)	*p*–Value
Modified FI2 Score (0–10)	RF	6.3 ± 3.2	10.0	0.048
	TA	4.7 ± 2.7		0.001
Age, in years	54.9 ± 17.2		50.5 ± 8.4	0.755
Sex, male, n (%)	5 (62.5)		6 (100)	0.282
Admission diagnosis, n (%)				1.000
Chest trauma	3 (37.5)		1 (16.7)	
Abdominal blunt trauma	1 (12.5)		3 (50.0)	
Both (chest and abdomen)	4 (50.0)		2 (33.3)	
Severity of trauma (ISS)	22.5 ± 7.6		19.0 ± 5.9	0.491
ICU Stay (d)	16.8 ± 19.0		24.7 ± 23.0	0.755
Mechanical ventilation (d)	12.0 ± 18.0		18.3 ± 16.7	0.573
MRC-SS (0–60)	52.3 ± 9.0		55.0 ± 3.6	0.755
Grip strength (kg, dominant hand)	21.5 ± 12.2		27.2 ± 8.9	0.240

Data are expressed as mean ± standard deviation; *p*-value from a Mann–Whitney test.; FI2, Functional Index-2; RF, Rectus Femoris; TA, Tibialis Anterior; ISS, Injury Severity Score; ICU, Intensive Care Unit; MRC-SS, Medical Research Council sum score.

**Table 2 healthcare-11-00053-t002:** Results of muscle endurance testing.

		Fatigue Patients (n = 8)	Non-Fatigue Patients (n = 6)	*p*–Value
Surface EMG	RF MVC (uV)	55.0 ± 14.8	154.9 ± 88.0	0.020 *
	RF %MVC (%)	42.7 ± 21.5	47.0 ± 12.0	0.755
	RF MDF (Hz)	66.3 ± 7.8	67.8 ± 9.5	1.000
	RF FI (%)	3.9 ± 3.8	4.6 ± 6.8	0.639
	TA MVC (uV)	144.0 ± 49.2	208.8 ± 53.5	0.043 *
	TA %MVC (%)	42.9 ± 22.9	42.5 ± 14.4	1.000
	TA MDF (Hz)	82.0 ± 8.1	81.2 ± 5.9	0.589
	TA FI (%)	9.3 ± 7.3	5.4 ± 6.2	0.445
Ultrasound	RF 2nd Echogenicity	67.7 ± 12.5	59.8 ± 15.2	0.662
	RF Echogenicity Δ	4.97 ± 10.0	1.8 ± 15.9	0.755
	TA 2nd Echogenicity	71.0 ± 15.1	61.4 ± 10.0	0.228
	TA Echogenicity Δ	7.2 ± 10.1	−1.4 ± 7.3	0.081
Borg CR10 (0–10)		3.5 ± 2.1	2.0 ± 2.1	0.228

Data are expressed as mean ± standard deviation; * *p* < 0.05; EMG, Electromyography; Borg CR10, Borg Category Ratio-10); MDF, Median Frequency; MVC, Maximal Voluntary Contraction; FI, Fatigue Index; RF, Rectus Femoris; TA, Tibialis Anterior.

## Data Availability

Not applicable.

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
