# Peer review of "Feasibility of Muscle Endurance Testing in Critically Ill Trauma Patients: A Pilot Study"

_healthcare, 2022, doi:10.3390/healthcare11010053_

Round 1

Reviewer 1 Report

Dear Authors,

Your study Feasibility of muscle endurance testing in critically ill trauma patients

 is interesting, it relates to a current problem, but have great limitations. The study needs some clarifications.

Intensive care unit-acquired weakness is a common and highly serious neuromuscular complication in critically ill patients, significantly impacting rehabilitation, and increasing both morbidity and mortality.

The first sentence in the abstract contains an error and the statement: “Neuromuscular weakness in critically ill is incentive care-unit acquired weakness (ICUAW)…” is somewhat simplified.

I have a few comments:

Primarily, the number of evaluated patients (14 patients in this study, eight in the fatigue group and six in the non-fatigue) is definitely not sufficient to express a conclusion about the suitability of using a certain technique to evaluate ICUAW, which is in itself quite problematic (CT, MRI, DEXA, ultrasound, dynamometer, bioimpedance, biochemistry – Creatinine, Cystatine).

If I understood correctly, 8 patients were included in the fatigue group when they were unable to complete the 3-min and FI2. Thus, it was feasible only in 6 patients of the non-fatigue group.

In addition, the evaluated method “The Functional Index-2 (FI2)” had to be modified in the study so that it could be used in trauma patients due to the limitations, and it only evaluates the endurance of the lower limbs. It was explained by the presence of cannulae and the limitation of upper limb movement but this carries the risk of not being suitable for use in otherwise limited patients (there exist many restrictions of movement due to various catheters and monitoring equipment, and this is generally more predominantly true for the upper extremity). It also requires a high degree of cooperation (Richmond Agitation-Sedation Scale from -2 to +2).

The biggest problem is ICUAW in the critically ill. Critical illness is characterized by a systemic inflammatory response leading to metabolic stress with the development of multiorgan dysfunction syndrome. Muscle dysfunction is an important part of this syndrome, and the degree of catabolism corresponds to the severity of the condition.

From a practical point of view, in critically ill patients with multiorgan dysfunction, this test recommended by the given study is practically difficult to perform. This study did not convince me from the six evaluated patients that it is an applicable tool for monitoring functional levels and recovery in critically ill patients. Although the finding of usability of the sEMG monitoring is interesting. The widely used ultrasound in this study was not sufficient but again, we evaluate the method from the results of 14 patients (groups of 8 vs. 6 patients).

This study did not convince me of its conclusion.

Reviewer 2 Report

Thanks for giving me the opportunity to review this manuscript.

The authors present a study in which they explored the feasibility of assessing muscle quality and endurance in trauma ICU patients.This study must be seen as a pilot study with only a small number of fourteen included patients. These patients are further divided into two groups of fatigue patients and non-fatigue patients. Only one patient met the ICUAW criteria.

Most of the authors´ statements should therefore be interpreted cautiously in view of the sparse data.

The introduction is kept extremely short and should urgently be expanded to include further preparatory work. A red thread would also be helpful in the course of the manuscript and should ideally follow the same structure  in all sections. I suggest starting the discussion section with the most important result.

In principle, this study is well thought out, shows a very homogeneous study collective and, for example, sonographic examination is also rather solidly performed by only one examiner with five years of experience. Therefore the study could possibly help to find a standardized procedure in future for evaluating muscle quantity and quality status in critically ill (question of definition, actually injured - there is also a lot of preparatory work in this regard) after including a larger number of patients.

Round 2

Reviewer 1 Report

Review 2nd

Feasibility of muscle endurance testing in critically ill trauma patients

Intensive care unit-acquired weakness is a serious neuromuscular complication in critically ill patients.

Previous comments:

1.The first sentence in the abstract contains an error and the statement: “Neuromuscular weakness in critically ill is incentive care-unit acquired weakness (ICUAW)…” is somewhat simplified.

The abstract is modified, improved, but there reamins a mistake in the very first sentence, ICU-AW stands for Intensive care unit acquired weakness not incentive

2. Primarily, the number of evaluated patients (14 patients in this study), is definitely not sufficient to express a conclusion about the suitability of using a certain technique to evaluate ICUAW, which is in itself quite problematic (CT, MRI, DEXA, ultrasound, dynamometer, bioimpedance, biochemistry – Creatinine, Cystatine).

The authors have changed the title to the pilot study, but we cannot count on statistically significant conclusions.

And I insist on the conclusion that this pilot study did not convince me from the six evaluated patients that it is an applicable tool for monitoring functional levels and recovery in critically ill patients.

But it is a pilot project in the difficult monitoring of ICU-AW, which describes the possibilities of investigation, which I evaluate positively.

3. In some places, abbreviations such as ROM are missing

Reviewer 2 Report

The authors made some corrections in their manuscript according to the recommendations of the review. This improved the form and understandability of the manuscript. The main weaknesses of the study remain the same as detailed in the first review: A modified test (FI2) is performed on a group of patients who essentially do not meet the diagnostic criteria for the disease to be investigated (ICUAW). Even if some interesting points are raised, the limitations are still too serious, even under the designation as a pilot study now, the conclusions can hardly be derived from the data mentioned, so that significant changes still urgently have to be made.
